

# A multimedia art interaction model for motion recognition based on data-driven model

Zhen Wang

Henan Economy and Trade Vocational College, Zhengzhou, Henan, China

## ABSTRACT

Ergonomics is a relatively important part of user experience in multimedia art design. The study aims to improve human–computer interaction efficiency using a data-driven neural network model combined with video data and wearable devices to achieve high-precision human motion recognition. Firstly, the human motion skeleton information is extracted based on video information through the OpenPose framework, and the motion characteristics of each joint of the human body are calculated. Then, use the inertia data of the wearable bracelet to calculate the exercise intensity. At last, the two kinds of information are sent to recurrent neural networks (RNN) together to achieve high-precision human motion recognition. The innovative contribution of the article is to establish a multimodal fusion model for human activity recognition. The experimental results show that the recognition precision of the proposed method reaches 97.85%, much higher than the backpropagation neural network (BPNN) and K-nearest neighbor (KNN), whose precision is 94.35% and 90.12%, respectively. The method's superior performance convinces us that the model can provide strong technical support for the interaction design of multimedia art.

# INTRODUCTION

As a medium, multimedia art encompasses creating sensory experiences through the integration of words, pictures, influences, sounds, and animations. With the continuous advancement of science and technology, the concepts of digitalization and informatization have become deeply ingrained in our lives, leading to the evolution and transformation of the definition of multimedia (*Chen, 2012*). The emergence of computers, mobile phones, various portable mobile devices, and their software has given rise to a new multimedia era, stimulating our senses in diverse ways. As tools and techniques of artistic design continually evolve, the concept of multimedia art also undergoes constant change. Currently, multimedia art can be categorized into various types, including video art, multimedia art, network art, remote communication art, and human–computer interface art. These art forms have forged closer connections with people, creating immersive and interactive experiences. In multimedia art design, human–computer interaction is advancing towards multichannel, multi-input, and multi-dimensional tracking recognition. This encompasses incorporating technologies such as micro-expression recognition, voice recognition, and

Corresponding author
Zhen Wang, wzjean05@163.com

virtual reality. These technologies enable artists and designers to explore new avenues of expression, engaging viewers in novel and captivating ways. Ergonomics plays a crucial role in multimedia art interaction, and its importance cannot be ignored. It involves designing and constructing multimedia art interaction systems to ensure that users (including artists, viewers, and participants) can interact with digital content in a comfortable, efficient, and secure environment. Ergonomics helps to create a more user-friendly and easy-to-use multimedia art interaction experience. By considering users' cognitive, perceptual, and behavioral characteristics, we can design user interfaces, interaction methods, and control systems to make it easier for artists and audiences to participate in artistic creation or experience. However, overcoming technological complexity, improving user-friendliness, and maintaining innovation are key challenges in this field, requiring continuous research and efforts to achieve a perfect multimedia art interaction experience.

The evolution of multimedia art in conjunction with technology presents possibilities for artistic expression and audience engagement. As technological advancements continue to unfold, multimedia art is poised to develop further, offering even more innovative and immersive experiences that challenge the boundaries between art, technology, and human perception. Certain distinct and relatively independent properties have emerged in multimedia design as the discipline has evolved. These include interactivity, multimedia unity, sensory coordination, digital media, and multi-dimensional space. While keeping people at the center of the design process, it is crucial to consider and incorporate these characteristics closely. The key elements in multimedia design revolve around stimulating the human senses, encompassing graphics, animation, video, audio, text format, and interactive components. With the rapid development of human–computer technology, art that places people at the forefront continues to emerge. Accurately discerning people's behavioral intentions and ensuring seamless human–computer interaction are prerequisites for the advancement of multimedia technology. By comprehending and catering to the needs and preferences of users, multimedia designers can create engaging and interactive experiences that captivate and deeply resonate with audiences.

With the relentless advancement of science and technology and the ever-improving computing power of computers, human-body recognition technology based on video technology has experienced remarkable growth within multimedia. From film production to virtual reality (VR) games, we constantly enrich our daily lives. At a deeper level, developing these technologies benefits people's artificial intelligence and deep learning development. In the image field, the convolutional neural network is the most commonly used depth learning algorithm, and classical neural network structures such as AlexNet, VGGNet, GoogleNet, ResNet, and DenseNet have been derived (*Simonyan & Zisserman, 2014*; *He et al., 2016*; *Wang, Qiao & Tang, 2015*). Compared with static images, video contains time-sequence features. A multichannel neural network is a common deep-learning algorithm in motion recognition using video. Two common streams include trajectory pooled deep cooperative descriptors, two streams Super-Resolution Convolutional Neural Network (SR-CNN), Temporal Segment Network (TSN), *etc.* (*Wang et al., 2016*; *Cao et al., 2017*). These methods greatly improve the processing efficiency of video stream generation. In the multimedia art design process, there is often no need

for redundant information in the picture and only need to recognize and classify human motion features.

OpenPose is established based on this requirement (*Cao et al., 2017*). The project uses a convolutional neural network and a supervised learning method. With Caffe as the development framework, the idea of going from bottom to top is adopted. First, human joint points are found in the picture, and then these joint points are used to splice adults. Finally, problems from top to bottom are decoupled to achieve a high-precision estimation of human posture (*Chen, Wang & Peng, 2018*). At present, this method can detect joint point information in the image, including human trunk nodes, toes, hands, and face nodes. In essence, the method is still a data-driven model, and the overall accuracy of the algorithm is improved through supervised learning training of a large amount of data (*Wang, Kläser & Schmid, 2013*). The core principles of data-driven neural network models for human motion recognition lie in feature extraction and neural network architecture. Firstly, the model can capture important information about motion by extracting key features such as acceleration, angular velocity, and joint angle from sensor data or video images. Then, these features are input into a neural network that can recognize and classify different human movements through multiple learning and weight adjustment levels. The advantage of neural network models is their ability to automatically learn and adapt to various complex motion patterns, making them a key tool for high-precision human motion recognition. We can achieve more accurate and robust motion recognition by continuously optimizing neural networks' structure and training process, providing broader application prospects for multimedia art interaction and other fields.

Using the inertial sensor in the bracelet to capture the pose can compensate for the image problem's occlusion and better solve the privacy problem. The inertial sensors bound to the key nodes of the body can also be used to achieve high-precision recognition of human posture using generalized discriminant analysis (GDA) (*Madgwick, Harrison & Vaidyanathan, 2011*), Extended Kalman Filter (EKF) (*Li et al., 2019*), Mahony (*Qiu et al., 2016*), and other pose analysis algorithms. However, relying solely on inertial equipment can lead to algorithm inaccuracies due to the influence of binding positions and severe jitter. Consequently, combining two high-precision human action technologies will significantly enhance motion recognition accuracy and efficiency in human–computer interactions.

Motion analysis related to the medical process has been a hotspot for motion recognition research these years. *Esteva et al. (2019)* reviewed the application of deep learning in the heal care. *Sutton et al. (2020)* described the support of intelligent systems in clinical risks. Motion recognition using hybrid sensors has also been investigated recently. For example, *Xu, Qian & Zhao (2018)* proposed a human move method indoors using inertial measurement unit (IMU) sensors. *Guo et al. (2018)* realized the motion recognition suing WiFi and Kinect information. *Wei, Chopada & Kehtarnavaz (2020)* combined video and inertial information to recoginize the motion with CNN model.These sensor-based motion recognition researches can be summarized in Table 1. It can be found that multi-sensors are gradually employed in motion recognition to improve accuracy.

This study aims at the requirement of efficient human–computer interaction in multimedia art design. It uses the fusion of video data and wearable devices to form an

| Table 1 The multi-sensor based motion recognition. | |
|---|---|
| **Research** | **Sensor type** |
| *Xu, Qian & Zhao (2018)* | IMU and WIFI |
| *Guo et al. (2018)* | Kinect and WIFI |
| *Wei, Chopada & Kehtarnavaz (2020)* | IMU and Video |

adaptive decision-making layer method to solve the problem of poor robustness of single information recognition. The data-driven OpenPose method based on a convolutional neural network extracts human motion features. It combines them with the inertial sensor features of wearable device terminals to form joint motion features. On this basis, the decision-making layer uses another data-driven RNN model to finally complete the decision fusion of video and inertial data and the high-precision recognition of human actions.

By incorporating data from wearable devices, the researchers aim to gain additional insights into the user's exercise intensity and physical activity while interacting with multimedia art. This multimodal fusion approach helps create a more comprehensive understanding of human motion, potentially leading to a more accurate and refined human motion recognition system.

The article's novelty is not solely on the motion skeleton extraction but rather on integrating and synergizing two data sources (video and wearable device data) using a data-driven neural network model (RNN). This combination of data allows for a more robust and precise human motion recognition system, ultimately contributing to a more efficient and ergonomic user experience in multimedia art design.

The rest is organized as follows: 'Introduction' introduces the development and related works for the art interaction; 'Method' describes methods used and the model establishment; In 'Experiment Analysis and Results', the experiment result and analysis are given; 'Discussion' discusses the result; the Conclusion is presented at the end.

# METHOD

According to the development characteristics of the multimedia art design and the requirements of art interaction, combined with the existing data characteristics and application methods, the proposed human motion recognition method flow is shown in Fig. 1.

Combine the human motion data collected by an RGB network camera with the inertial information collected by wearable devices. After the fusion of the two features is completed, send the data into the data-driven model to complete the supervised model training and verify the data by dividing the data set to form a multimedia art human motion recognition model.

## The framework for openpsoe

In human–computer interaction, human motion recognition plays a pivotal role, and extracting human motion features holds significant importance. However, when confronted

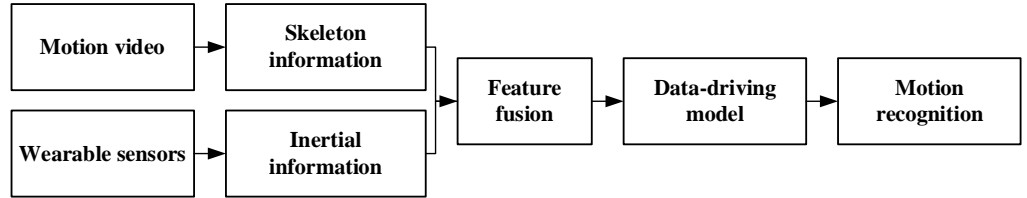

**Figure 1** **The framework for the multimedia art interaction motion recognition.** Combined with the existing data characteristics and application methods, the proposed human motion recognition method flow is shown.

with scenarios involving an uncertain number of people, occlusion of objects, or occlusion between individuals, these factors can considerably impact the recognition methodology based on image classification for human motion postures. Consequently, to address this challenge, the OpenPose neural network introduces the concept of part affinity fields (PAFs). Essentially, PAFs represent a set of two-dimensional vector fields that contain crucial information about the position and direction of the limbs. The degree of association between these joint points can be calculated by utilizing the positional information and PAFs associated with each joint point of the skeletal structure. This calculation enables the subsequent fusion of the human model, completing the process of assembling a comprehensive representation of the human body and its movements (*Shi, Wang & Zhao, 2022*).

OpenPose is a two-step framework that can be expressed by the Eqs. (1) and (2) (*Qiao, Wang & Jian, 2018*):

$$S^t = \rho^t(F, S^{t-1}, L^{t-1}) \tag{1}$$

$$L^t = \varphi^t(F, S^{t-1}, L^{t-1}). \tag{2}$$

St is the joint point confidence map output, and Lt represents the nodes relationship among different nodes, which can also be named as the weight coefficient. In Eqs. (1) and (2), the time step t are both more than 2. Considering the loss function, L2 loss function was employed for both branches. The loss in S and L can be shown as Eqs. (3) and (4):

$$\text{Loss}_s^t = \sum_{i=1}^{J} \sum W(p) \bullet \|S_i^t(p) - S_i^*(p)\|_2^2 \tag{3}$$

$$\text{Loss}_L^t = \sum_{j=1}^{J} \sum W(p) \bullet \|L_j^t(p) - L_j^*(p)\|_2^2 \tag{4}$$

$$L = \sum_{t=1}^{T} (f_s^t + f_L^t) \tag{5}$$

where superscript $^\star$ is the true value and the one without that represents the predicted value, p represents each pixel, W (p) means the point is missing at this point. Combining the loss from two steps, the overall loss is obtained in Eq. (5).

After the preliminary training of the network, confidence detection of the human body region is required. There are 19 key joint nodes in an image of the human body. For the k person, the confidence map of position j will be generated. xj, k are the positions of the kth individual part j in the figure. It is through Gaussian modeling to obtain $S_{j,k}^*(p)$ the thermal diagram of the pixel p in the image with respect to the body part j. The calculation formula is shown in Eq. (6). the maximum value of the response is taken for multiple responses generated.

$$S_{j,k}^*(p) = \exp(-\|p - x_{j,k}\|/\sigma^2). \tag{6}$$

On this basis, complete the skeleton node association based on the partial affinity field and realize the bipartite graph matching with the maximum edge weight value to achieve high-precision calculation of human motion posture. The projection relationship between two points in the limb is used to calculate the angle between each joint. The calculation can be represented by the following Eq. (7) as follows:

$$\theta_n = \arccos \frac{r_{i,j}^1 \bullet r_{i,k}^1 + r_{i,j}^2 \bullet r_{i,k}^2}{\sqrt{r_{i,j}^{12} + r_{i,j}^{22}} + \sqrt{r_{i,k}^{12} + r_{i,k}^{22}}}. \tag{7}$$

Among them, $r_{ij}$ represents the vector of the limb joint necklace starting from i to j. $\theta_n$ represents the angle between the $n$th limb and is formed by the shutdown calculation between the two limbs $r_{ij}$ and $r_{ik}$. Therefore, the original video stream data can be extracted through the OpenPose network and joint point data to obtain the data relationships between various joints, forming the angle features of the motion joints.

## Key motion feature extraction using wearable devices

The motion information acquisition of wearable devices mainly depends on the inertial sensors carried by the device itself. Inertial devices applied to wearable devices generally have low accuracy at low cost, and their accuracy is limited. Only time-domain signals are analyzed according to the actual situation of the collected signals. In this article, the sliding average method of the most used digital filtering method is adopted, and its calculation method is shown in Eq. (8) (*Wang et al., 2018*)

$$p_t = \frac{\sum_{i=1}^n (x_{t-i} + x_{t+i}) + x_t}{2n + 1}. \tag{8}$$

The window size directly affects the filtering result. If the window is too large, the filtering result will be smoother. However, the small window size may deviate from the true value to some extent and the large window may induce more noise.

As data filtered, the feature representing the motion should be extracted. In this article, the traditional statisticals like the mean, extre and skewness are all calculated. At the same time, the motion variance and intensity based on the sliding window is also employed as shown Eqs. (9) and (10)

$$\sigma_{a_i}^2 = \frac{1}{2w + 1} \sum_{j=i-w}^{j=i+w} \left(a_j - \bar{a}_j\right)^2 \tag{9}$$

$$M_{\text{in}} = \sqrt{a_x^2 + a_y^2 + a_z^2} \tag{10}$$

where $\sigma_{a_i}^2$ represents the sliding variance results of each axis of the three-axis acceleration, aj is the heart rate value at the j, w is the sliding window's width, and the mean value in the j point window. Equation (9) can be obtained by calculating the square sum of the three-axis accelerations in the inertial sensor, which can reflect the intensity change of motion with high accuracy to a certain extent (*Peng et al., 2021*).

## Motion recognition based on the data-driving model

The data-driven model can be close to the law of data according to the change of data itself; that is, the model can adapt to the situation and change according to different data changes. Most of the traditional machine learning methods belong to this category, and the model training is completed by collecting data. However, in multimedia human–computer interaction, all kinds of actions are often not independent. So, it is necessary to refer to the actions in the previous stage, that is, consider the continuity of the time series to complete. Therefore, although traditional machine learning algorithms such as K-nearest neighbor (KNN), support vector machine (SVM), C4.5 (one of the decision tree methods), *etc.*, can achieve classification in time step, they do not have time consistency. Estimating the previous state is difficult, which greatly reduces the experimental results. Therefore, the cyclic neural network is needed for the sequence with time continuity. The mathematical expressions of traditional neural networks and recurrent neural networks (RNN) can be expressed by Eqs. (11) and (12), respectively (*Wagstaff & Kelly, 2018*):

$$h_t = \sigma(\omega_{x_t} x_t + b) \tag{11}$$

$$h_t = \sigma(\omega_{x_t} x_t + \omega_{h_t} h_{t-1} + b). \tag{12}$$

It can be found from Eqs. (11) and (12) that recurrent neural network (RNN) adds the input state ht-1 of the previous time in the function input process, which allows it to remember the output state of the previous time, making it an advantage in processing time series. The fundamental characteristic that sets RNNs apart from other neural networks is their ability to maintain a hidden state or memory of past inputs while processing the current input. This hidden state allows RNNs to capture temporal dependencies and learn patterns over time. RNNs use the same set of weights across all time steps. This shared weight structure enables them to learn and recognize patterns in sequential data efficiently. Therefore, this article completes recognizing human motion sequences through the cycle time network RNN.

## EXPERIMENT ANALYSIS AND RESULTS

### Human skeleton information

In this article, to meet the interaction needs in multimedia art design, when designing experiments, first, let volunteers wear a bracelet to collect inertial information on their hands, then let them sit on a stool and remain still for a period, and then stand up to complete the lifting of their left hand, right hand and hands in turn. Three volunteers were recruited for the experiment, and each experiment was repeated five times. The

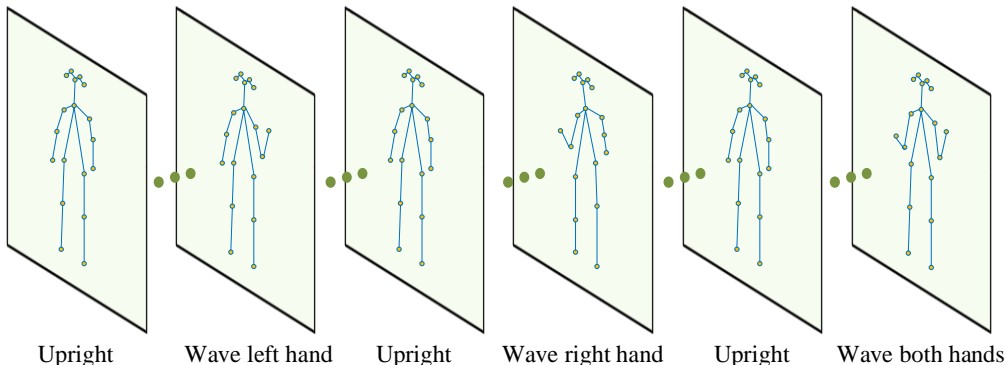

| Upright | Wave left hand | Upright | Wave right hand | Upright | Wave both hands |

**Figure 2** **Human skeleton information obtained from the video.** The skeleton information extracted through OpenPose is shown in Fig. 2.

data's safety should also be ensured when capturing the volunteers' video. Hence, in this article, the video is only used for scientific use and the consent is also acquired from the volunteers (*Haider et al., 2021*). After capturing the video, the subject-specific training was selected; two subjects were used to train the model; one was the test.

The camera used in the experiment is an RGB webcam. The skeleton information extracted through OpenPose is shown in Fig. 2, which is limited by the height, obesity, and thinness of different people. In addition to referring to Eq. (6), if extracting joint information, the skeleton is adjusted before using it to calculate joint angles to improve the calculation accuracy of each joint angle.

## Inertial information from wearable device

For the inertial signal, given the characteristics of the current equipment, this study collected the hand motion information of the experimental object. The 3D acceleration signal of the hand during the experiment is extracted. After extraction, the noise in the movement is removed using mean filtering. One of the simplest ways to gauge exercise intensity is by calculating the magnitude of acceleration. Higher acceleration values generally correspond to more intense physical activity. For example, running or vigorous exercise typically produces higher peak acceleration values than walking or sitting. The three-axis acceleration information obtained is shown in Fig. 3. The characteristics of the collected acceleration signal are different, which is crucial for signal recognition in the next stage.

In data collection, we first had the volunteers who received the test stand still and conducted a familiarity test after wearing it according to their movement mode. After confirming that wearing the bracelet did not affect their movement, we conducted an experimental test. The experimental subjects performed the specified sequence of activities in the designated area with videos according to the requirements, thus completing the data collection.

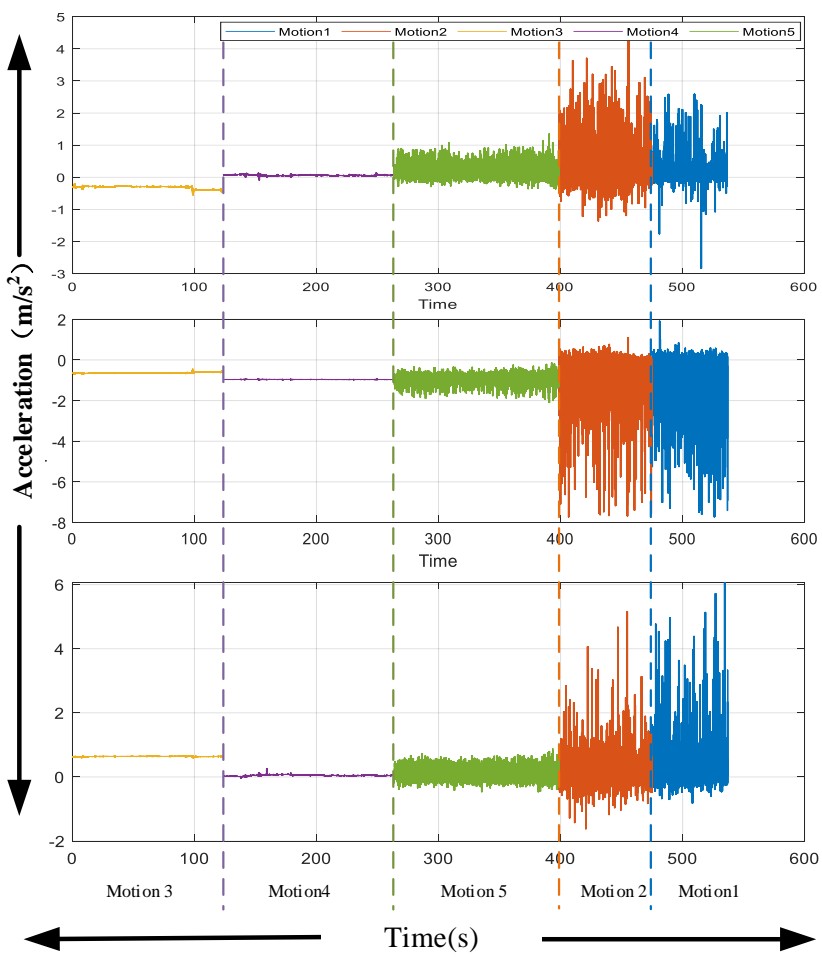

**Figure 3  Acceleration from wearable devices.** The three-axis acceleration information obtained is shown in Fig. 3.

## Motion recognition results

To more accurately explain the recognition results based on the data-driven model in this article, the recognition is evaluated by precisopm. Recall and F1 score, which can be represented from Eqs. (13)–(15):

$$P = \frac{TP}{TP + FP} \tag{13}$$

$$R = \frac{TP}{TP + FN} \tag{14}$$

$$F1 = \frac{2 \times P \times R}{P + R}. \tag{15}$$

TP represents true positive; FP indicates false positive; FN is the false negative. For this reason, the evaluation results are assessed by these three indicators.

**Table 2  Results of the motion recognition.**

|  | Sitting | Standing | Left wave | Right wave | Both wave | Mean |
|---|---|---|---|---|---|---|
| P | 98.86% | 99.12% | 96.32% | 96.45% | 98.5% | 97.85% |
| R | 97.44% | 97.67% | 94.23% | 98.78% | 94.03% | 96.43% |
| F1 score | 98.14% | 98.39% | 95.26% | 97.60% | 96.21% | 97.13% |

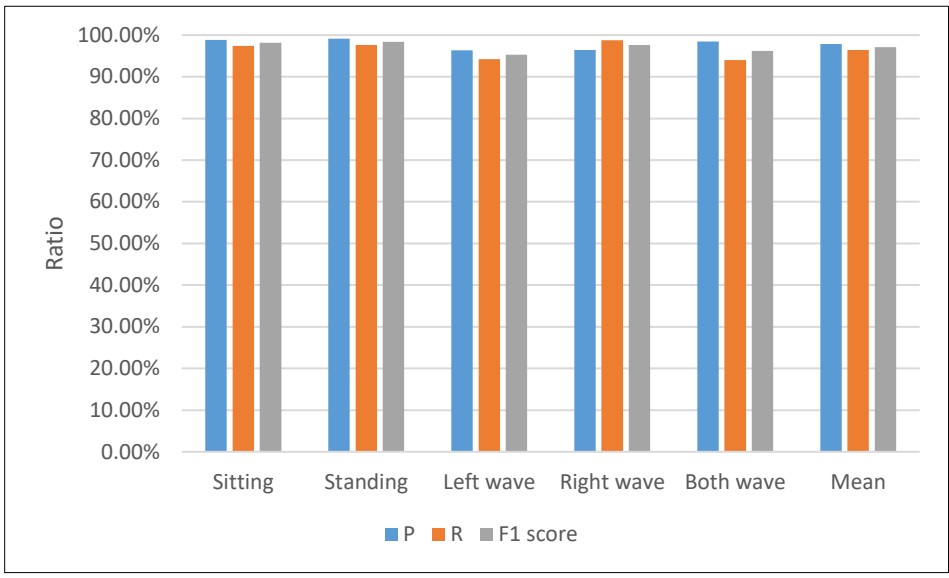

**Figure 4  The result for the motion recognition among different indexes.**

This study recognizes human actions through the extracted joint angle information and hand posture acceleration and motion intensity information. The recognition results are shown in Table 2 and Fig. 4.

The proficiency of the approach in this manuscript is readily discernible from Table 1, demonstrating exemplary performance across accuracy, recall, and various other facets. Simultaneously, a sense of equilibrium pervades the recognition rates of individual data points, with each category attaining commendable recognition levels.

For a more comprehensive elucidation of the experimental findings, this study employs models predicated on diverse experimental subjects during the training phase, with the corresponding outcomes depicted in Fig. 5.

The model also has good recognition results in the recognition of specific objects. Among them, subject 2 has the highest accuracy rate of recognition results because there are no additional small actions in the movement process. It faces the camera in the process, resulting in a more accurate extraction of joint angles. In contrast, subject 1 has a more general recognition result because compared with the other two objects, there are significant differences between each group of actions during the experiment. For example, the speed of arm lifting is sometimes fast or slow, leading to low recognition accuracy, which needs to be focused on in future experiments. However, no matter the overall or

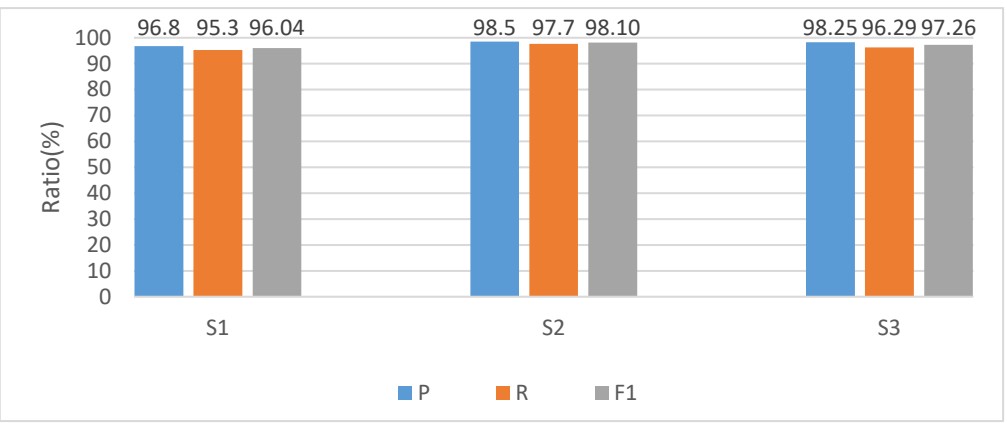

**Figure 5** Recognition results among different subjects.

single-person recognition rate, the model shows excellent recognition results with more than 95% accuracy.

## The application in the art interaction

Simultaneously, to enhance the elucidation of the proposed model's practical utility in art, this study employs the model to assess the action sequences inherent to multimedia technology interaction. Field trials were conducted with a local video special effects production company, aligning with their specific requirements. The company successfully executed action recognition within a video clip, showcasing the results in Fig. 6.

This study selected three objects for actual testing. Due to the small amount of data in the actual testing, we adopted the experimental method of leaving one and tested the three indicators mentioned above. Figure 6 shows the RNN, BP neural network, KNN, CNN and SVM models from left to right. The RNN model selected in this study is superior to the traditional BP neural network model in terms of accuracy or recall. At the same time, it can be found the performance of KNN is poor because of such relatively continuous action recognition. Through actual comparison, it is found that the identification efficiency of this article is very high and has been praised by local companies.

## DISCUSSION

Human–computer interaction plays a pivotal role in multimedia art design, serving as a central focus in multimedia technology research. Its primary objective is translating human gestures and actions into corresponding interactions within virtual environments. Whether encompassing dance games, fitness applications, sound restoration, or film and television recording, all these pursuits underscore the paramount importance of human–computer interaction in multimedia art design. Within this context, human conduct is the foundational element for diverse forms of artistic expression (*Xiao & Shah, 2005*). Precisely discerning human action intentions is paramount in achieving precision in reproduction and enabling further processing. This article undertakes an exhaustive

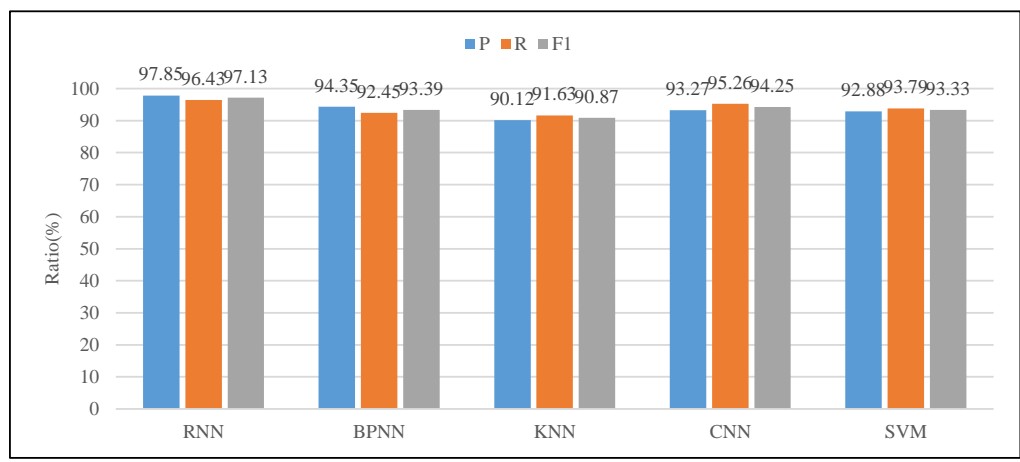

**Figure 6** **Results of different classification methods.**

examination of human–computer interaction in multimedia art design, exploring the application of OpenPose technology (*Lina & Ding, 2020*). This technology captures essential information about the human skeletal structure, forming a critical feature for human motion classification.

Additionally, addressing challenges such as unstable network connections, camera-related data transmission issues, and partial occlusion problems leading to data deviations is essential in practical applications. To overcome these challenges, this article proposes the integration of wearable bracelet devices, which act as supplementary sources of information and contribute to the recognition of multimodal data. Combining these approaches presents a comprehensive framework for human–computer interaction in multimedia art design. Many benefits can be achieved by integrating inertial information and video information for action recognition. Firstly, inertial information (such as acceleration and gyroscope data) can capture body motion and posture changes but may not provide sufficient detailed context. Video information can provide rich visual content, including the environment and human body position, to better understand the execution of actions. Secondly, data fusion can improve the accuracy and robustness of action recognition. Sometimes, relying solely on inertial or video information, such as lighting conditions or sensor errors, may be limited. Integrating the two can compensate for their limitations and provide more reliable results. In addition, data fusion can also achieve multimodal action recognition, which means that more types of actions can be identified or more details can be provided. This has potential applications in health monitoring, exercise training, and virtual reality.

In recent years, 3D convolutional neural network using Two Stream has also attracted more and more researchers' attention in the field of human motion recognition of video images applied separately, such as Slow Fast Network, whose main idea is to enhance the representation ability of features by inputting video frames with different sampling frequencies (*Farooq et al., 2022*). By adding the NN (Non-local neural network) attention mechanism to SlowFastNetwork (*Soomro, Zamir & Shah, 2012*), the network can achieve

**Table 3  The comparison among different datasets.** To compare the methods proposed, we employed two open datasets UTD-MHAD and C-MHAD. We choose the video data and inertial data collected on the wrist to imitate the data in our experiment and the results are shown in Table 3.

| Dataset | Precision | Method |
| --- | --- | --- |
| UTD-MHAD | 96.43% | Proposed |
| | 93.01% | *Wei, Chopada & Kehtarnavaz (2020)* |
| | 90.20% | *Chen, Jafari & Kehtarnavaz (2015)* |
| C-MHAD | 97.25% | Proposed |
| | 91.27% | *Wei, Chopada & Kehtarnavaz (2020)* |

the best recognition accuracy on UCF101 and Sports datasets. Video is composed of spatial information and temporal information. The convolutional neural and cyclic neural networks can be used to learn the spatiotemporal dimension characteristics of the video. Firstly, the spatial feature information of each frame is extracted from the video stream information through the convolutional neural network. Then, the spatial feature information of each frame is input into the cyclic neural network to remove the time dimension features between frames. Although there is much research on video action recognition algorithms based on deep learning, they often have only one output tag or only one person in the video. Although this can simplify the algorithm process and improve the accuracy of the algorithm, it is far from reality.

Moreover, the massive volume of the model also puts higher requirements for the actual deployment side. Therefore, this article selects the current open-source project OpenPose, focusing only on bone information and motion angle information, which can significantly reduce the difficulty of subsequent model design and improve the recognition rate of the model to a certain extent. But suppose it is based on inertial sensor information alone because of its comfort and flexibility of movement. In that case, it cannot be worn in a large area, which leads to the loss of information, and it is challenging to achieve high-precision action recognition. At this stage, in multimedia interaction, human–computer interaction is often conducted through handheld devices. Therefore, the combination of handheld devices and image data can significantly expand the scalability of the method proposed in this article. To compare the plans proposed, we employed two open datasets, UTD-MHAD and C-MHAD. We chose the video data and inertial data collected on the wrist to imitate the data in our experiment and the results are shown in Table 3.

As the March of science and technology continues unabated and algorithms undergo constant evolution, the future of multimedia interaction design must embrace the concept of device-free interaction. This visionary approach seeks to achieve the most genuine mode of interaction by endowing environments with the ability to perceive users' presence without reliance on physical devices. Through the astute utilization of full-space intelligent scenes and projection technology, the ultimate objective is to elevate the overall interaction experience (*Aljuaid et al., 2021*). Moreover, the essence of interactive information design lies in its simplification and flattening, which augments interaction efficiency. Users can effortlessly navigate through and engage with multimedia content by streamlining the design.

Additionally, establishing a comprehensive full-scene space intelligent environment accentuates the imperative of achieving realtime display across multiple platforms and devices. Only through the seamless integration of various platforms and devices can we truly elevate the human–computer interaction experience and realize the true potential of multimedia art design in a world that thrives on multimedia integration. Consequently, this would bestow greater convenience and elevate the quality of life in individuals' daily activities and creative pursuits.

The vanguard of future multimedia interaction design aspires to deliver unparalleled authenticity, immersion, and intelligence in user experiences through the synergistic deployment of advanced perception technologies, intelligent voice assistants, and the ubiquitous application of IoT. These momentous strides will bestow greater convenience and offer high-quality services that enrich individuals' daily lives and creative pursuits.

## CONCLUSION

The essence of art interaction design revolves around the intricate challenge of human–computer interaction within the domain of multimedia art creation. This article constitutes a pioneering effort, delving deeply into high-precision human motion recognition within the context of motion capture for film and television art. It accomplishes this feat by employing a cutting-edge multi-mode information fusion approach that synergistically amalgamates video and wearable technology realms. To achieve this groundbreaking feat, the authors harness the formidable capabilities of the OpenPose framework. This technology is instrumental in extracting intricate human skeleton information, a pivotal step that underpins the meticulous computation and replication of joint angles through a combination of data normalization and the meticulous design of complete limb angle features. In tandem, a wearable device ingeniously affixed to the hand captures invaluable motion inertia information, which forms the bedrock of motion intensity features when coupled with three-axis acceleration data. These motion intensity features are seamlessly amalgamated with the joint motion features meticulously derived from images, and together, they are channeled into a data-driven model founded on RNN. The result is a state-of-the-art system that achieves remarkable high-precision human motion recognition, as evidenced by the commendable results, showcasing an average recognition accuracy of 98.75%. This noteworthy precision significantly advances the field of human motion recognition, playing a pivotal role in the human–computer interaction process within multimedia art. In essence, this work represents a groundbreaking and pioneering study in motion recognition for multimedia interaction, pushing the boundaries of what is achievable in the convergence of technology and artistic expression. In our future endeavors, we are committed to expanding the boundaries of motion recognition technology by integrating an even more comprehensive array of data sources and employing advanced methodologies. We aspire to enhance the model's performance, tackle challenges associated with missing data instances, and refine our ability to discern subtle differences between similar motions. Furthermore, we aim to explore more sophisticated feature extraction methods, including incorporating contextual information from multiple sensors and using

temporal modeling to capture nuanced patterns within motions. This approach will allow us to achieve a higher level of granularity in motion recognition, enabling our model to distinguish between highly similar actions with greater precision.

### Funding
The author received no specific funding for this study.

### Competing Interests
The authors declare there are no competing interests.

### Author Contributions
- Zhen Wang conceived and designed the experiments, performed the experiments, analyzed the data, performed the computation work, prepared figures and/or tables, authored or reviewed drafts of the article, and approved the final draft.

### Data Availability
Data: https://www.zenodo.org/record/804402. 10.5281/zenodo.7131715.

Code files have been uploaded as Supplementary Files.

### Supplemental Information
Supplemental information for this article can be found online at http://dx.doi.org/10.7717/peerj-cs.1662#supplemental-information.

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
