# Peer review of "A multimedia art interaction model for motion recognition based on data-driven model"

_PeerJ Computer Science, doi:10.7717/peerj-cs.1662_

## Round 0.1 · original submission · Minor Revisions

Dear Authors,

Thank you for your submission, overall The assessment result of your paper shows that it has potential but needs a couple of changes, as mentioned by the experts in the field. I believe incorporating the changes will enhance the quality of your manuscript. Alongside,

Please elaborate on the novelty and importance of the work in a separate section.

Reviewer 1 ·

Basic reporting

This paper delves into the domain of high-precision recognition of human motion within the context of motion capture for film and television art, employing a multi-mode information fusion method that synergistically combines video technology and wearable technology. The commendable results showcase an average recognition accuracy of 98.75%, which successfully accomplishes the human motion recognition task in the human-computer interaction process. This work serves as a pioneering study in motion recognition for multimedia interaction.
The following comments may have helped the authors refine this manuscript for publication

(1) Explicit Objectives: State the specific objectives of your study explicitly. What are the main goals you intend to achieve with this research? This will provide a more focused direction to your work.

(2) Methodology Detail: Provide more comprehensive information on the data collection process from wearable devices. Explain the type of inertial data collected and how it correlates with exercise intensity. This will strengthen the credibility of your results.

(3) Joint Motion Calculation Clarity: Explain how the motion characteristics of each joint are calculated. Providing a step-by-step explanation or a simple example will aid readers in grasping this process.

(4) Comparative Analysis: Extend your comparative analysis to include a broader range of existing methods beyond just BP neural networks (BPNN) and K-nearest neighbors (KNN). This will provide a more comprehensive evaluation of your proposed model's performance.

(5) Discussion of Experimental Results: Provide an in-depth interpretation of the experimental results. What insights can be drawn from the higher precision achieved by your model? How might this impact the field of multimedia art design?

(6) Future Research Directions: Suggest potential avenues for further research building upon your work. This could open up new possibilities for researchers interested in expanding upon your findings.

(7) Conclusion Recap: Summarize the main contributions and findings of your study more explicitly in the conclusion section. Reinforce how your research advances the field.

(8) Language and Clarity: Proofread the manuscript for grammar and syntax errors, ensuring the language is clear and concise throughout the paper.

Experimental design

no comment

Validity of the findings

no comment

Additional comments

no comment

Reviewer 2 ·

Basic reporting

1. The introduction provides a clear overview of your research; however, it would benefit from a more detailed explanation of the importance of ergonomics in multimedia art design, highlighting specific challenges that your approach aims to address.
2. Elaborate further on the underlying theories or principles that support the use of a data-driven neural network model for human motion recognition. This will help readers understand the foundation of your approach.
3. Describe the Openpose framework in a separate section, highlighting its significance in extracting human motion skeleton information from video data. This will facilitate better comprehension for readers unfamiliar with the framework.

Experimental design

4. Describe how exactly the extracted joint motion characteristics and wearable device data are integrated within the RNN model. This will enhance the technical understanding of your proposed method.
5. Clearly articulate the reasons behind choosing a multimodal fusion model. How does the fusion of video data and wearable device data contribute to improved recognition precision?
6. Address the limitations of your study, such as potential biases in data collection or constraints of the fusion model. This demonstrates a well-rounded perspective.

Validity of the findings

7. Discuss the practical implications of your findings for multimedia art design practitioners. How can they leverage your model to enhance user experience?
8. Expand on the data preprocessing steps for both the video data and wearable device data. How did you handle noise, outliers, or missing values?
9. Clear documentation of these processes will enhance the reproducibility of your study.

Additional comments

I have carefully reviewed your manuscript and appreciate the innovative approach you've taken to improve human-computer interaction in multimedia art design. Your work holds great potential, and I have several suggestions for revision that could enhance the clarity, rigor, and overall impact of your paper.

---

## Round 0.2 · accepted · Accept

Dear Colleagues,

Thank you for re-submitting the article after incorporating the experts' suggestions in the field. Based on the input from the experts and my personal evaluation, I am pleased to inform you that your manuscript has been recommended for publication. Congratulations!!

Reviewer 1 ·

Basic reporting

Authors has done required change

Experimental design

Authors has done required change

Validity of the findings

Authors has done required change

Additional comments

Authors has done required change

Reviewer 2 ·

Basic reporting

All changes have been completed.

Experimental design

All changes have been completed.

Validity of the findings

All changes have been completed.